# Vertical integration selection of Chinese pig industry chain under African swine fever - From the perspective of stable pig supply

**Gangyi Wang**[1,2], **Jingjing Wang**[1]*, **Siyu Chen**[1], **Chang´e Zhao**[1]

1 College of economics and management, Northeast Agricultural University, Harbin, Heilongjiang, China,
2 Chongqing Rural Revitalization Institute, Chongqing, China

* wangjingjing1115@neau.edu.cn

**Data Availability Statement:** All relevant data are within the paper and its Supporting Information files.

## Abstract

Vertical integration is conducive to the realization of complementary interests and sustainable development of pig industry. The outbreak of African swine fever (ASF) in 2018 has disrupted many activities along pig industry chain in China. The production capacity of breeding pigs has dropped rapidly, and the supply of pig is tight. The vertical integration of pig industry chain is the main driving force to ensure food supply. Based on the data of 12 listed pig companies from 2012 to 2019, we examine the ways and reasons for vertical integration of pig companies when external shocks increase by taking ASF as an example, breakpoint regression and Tobit model are used to analyze differences and determinants of the forward and backward integration of pig industry chain under ASF. The empirical results showed that the forward integration of the feed processing link and slaughter circulation link is higher than the backward integration. ASF had different effects on the vertical integration degree of each link. ASF promoted forward integration. The main factors have different influences on the vertical integration of pig industry in China. Forward integration increased mainly depending on previous asset specificity, legal system environment, market demand, and transaction frequency. The findings of the study imply that pig industry chain is taking the forward integration to cope with the ASF shock. The combination of feed link and breeding link reduces feed cost and ensures pig supply. Pig companies tend to reduce transaction costs by strengthening the control of downstream supply.

## Introduction

The pig industry is an important industry related to the national economy and people's livelihood in China. Pork production and consumption of China rank first in the world. The pig industry in China develops rapidly on the whole but the development of the pig industry chain before, during, and after production links is unbalanced. Some events impact the sustainable development of the pig industry through different periods. Any link affected by external

**Funding:** This research is supported by the National Social Science Foundation of China (Project number 22BJY084); the Humanities and Social Science Foundation of Ministry of education of China (Project number 21YJA790053); the Chongqing Talents Program in China; and "Scholars Program of Northeast Agricultural University" High-level Talents Project (JC2) in China. The funders had no role in study design, data collection and analysis, decision to publish, or preparation of the manuscript.

**Competing interests:** The authors have declared that no competing interests exist.

shocks will lead to huge fluctuations in the industry chain. With the rapid expansion of the overall scale of breeding and the improvement of breeding density, the threat of major diseases to the pig industry is also increasing. The outbreaks of African swine fever (ASF) in China affected many links in the pig industry chain. Although fewer than 200 outbreaks of ASF have occurred, the pig industry still faces profound challenges. The industrial chain is decoupled, the feed supply is seriously lagging, and it is difficult to guarantee the supply in breeding. There are "no pigs to slaughter" and "no pigs to transport" in slaughter and circulation. The pig industry has suffered huge losses. Starting in the third quarter of 2018, breeding sows have recorded a month-on-month decline of more than 1% for 10 consecutive months. Month-on-month change rates since 2019 have fallen by more than 10 percent, and the number of breeding sows in October was about 20.7 million, which was 37.92% lower than that in 2018. Pork consumption accounts for 63% of meat consumption in China.

Low industry concentration is the underlying cause of these problems, and it also brings great obstacles to solving these problems. Vertical integration of companies' internal industry chains has obvious comparative advantages in risk response. Spain, which has a similar pig industry structure to China, faced the same pig undersupply decades ago. There are low-scale and industrial concentrations of the pig industry in Spain. The number of slaughters fell under ASF. With the impetus of the Spanish government, the pig industry began to accelerate the development of vertical integration. Feed processing subjects take the initiative to develop backward integration into pig breeding and sign contracts with breeding subjects to achieve stable transactions, so the industrial concentration is further improved, and the scale is also rising rapidly. During this period, the number of scatter-feed households dropped by 110,000, and the number of slaughters on single farms increased from 122 to 467. As of 2017, the pig production of large-scale pig industry chains with vertical integration accounted for 65% of pig production in Spain.

In 2021, the "Opinions of the Ministry of Agriculture and Rural Affairs, the National Development and Reform Commission, the Ministry of Finance, the Ministry of Ecology and Environment, the Ministry of Commerce, and the China Banking and Insurance Regulatory Commission on Promoting the Sustainable and Healthy Development of the Hog Industry" pointed out that it is necessary to improve the stable production and pig supply. As the most populous country in the world, pork is the largest animal product in the food consumption of Chinese residents, accounting for 31% of the total animal food consumption. The pork supply is an important guarantee for the food supply in China. ASF has shocked the pig industry chain and exposed the pork supply to risks. From the perspective of practice, the pork supply needs to be completed by many companies. Listed pig companies occupy a prominent position in the development of the industry, which is mainly manifested in market concentration and product recognition, as well as their impact on the development of the pig industry and industrial policies. Vertical integration can increase the market concentration of companies and ensure food supply. Previous studies of pig industry chain have focused on pig breeding [1, 2]. When companies a members of the chain [3, 4], it can significantly affect the performance of the company and industry chain competitiveness [5–7].

ASF has disrupted pig industry chain [8], and the impacts on food supply even extend upward to the upstream production part of pig industry [9]. As a highly contagious disease, on the one hand, ASF has caused volatility pig market and hindered the sustainable and high-quality development of the pig industry in China [10–12]. ASF threatens the food supply and challenges the livelihoods of pig producers and other entities along the supply chain [13], and the total pig herd and pork production volume has decreased [14]. The drastic reduction in pork supply has led to a sharp increase in pork prices [15]. On the other hand, ASF has transformed pig industry in China into medium and large-scale farms, as well as better-

standardized production systems and biosecurity [14]. It can be seen that ASF has an impact on all aspects of pig industry chain, so it is necessary to find a way to ensure food supply.

Vertical integration is widely used by researchers because of its essential role in improving production and strengthening the connection between upstream and downstream of the industry chain [16, 17]. Although some studies have shown that vertical integration reduces production [18], most studies suggest that vertical integration can improve the performance of farmers [19]. Participation in written and verbal contracts in a vertical integration mechanism can increase the quality of the relationship between farmers and buyers, and positively affects the performance of farmers [20]. In addition, some countries increase pig industry chain value through vertical integration[21, 22], and vertical integration also effectively prevents disease in the pig industry [23]. Vertical integration can improve the performance of the pig industry, alleviate the impact of the epidemic on the industry chain, and ensure a stable food supply.

Previous studies provide a solid foundation for this paper. However, pig industry chain vertical integration needs to be further studied: First of all, the vertical integration of the industrial chain can improve the competitiveness of companies. The previous studies on the industry chain focus more on the performance or efficiency of the supply chain operation and have never explored what factors could affect the vertical integration of the industry chain. Secondly, previous studies did not consider the impact of the epidemic on the vertical integration of the pig industry chain. In theory, the epidemic will change the behavior of all entities in the industrial chain. It is necessary to explore the influence of ASF on the vertical integration of the industrial chain. At present, vertical integration of the pig industry chain will be an important direction to ensure food supply under the normalization of ASF in China. To deal with ASF or ensure food supply, the most important solution is to break the decoupling status of the pig industry chain. The vertical integration development promotion of the industry chain can stabilize the food system from the supply side. Therefore, this paper firstly analyzed the influence mechanism of transaction costs on the vertical integration of the industry chain from the theoretical level and then discussed the vertical integration degree of pig industry chain in China, the integration degree of each link, the obvious difference of transaction costs faced by the pig companies in different links and the determinants of vertical integration. Taking ASF as an example, this paper will explore the ways and reasons for vertical integration of pig companies when external shocks increase.

## Theoretical framework

The concepts of industry chain and supply chain are essentially consistent, aiming to coordinate the behaviors of various entities in the chain, improve the overall operating efficiency of the chain, and promote maximum utility. We define pig industry chain as a network structure composed of all functional links that have upstream and downstream relations with pig breeding under the relevant legal environment and resource environment. Pig industry chain is an open system, and the subjects independently select transaction models based on their interest relationships to improve their benefits. The external environment is constantly changing, which disturbs the organizational mode and transaction rules of the constituent elements in the chain.

The essence of industry chain integration is that companies break through boundaries, seek the best combination of resources in the industrial chain. And to reduce transaction costs, amend resource constraints by expanding the available resource space, and realize the transformation of resources from narrow sense to broad sense and from outside to inside.

Economists have explained the reasons of vertical integration from different perspectives. Vertical integration refers to the realization of a win-win situation for upstream and

downstream companies to improve efficiency and reduce costs through the optimization of the value chain, supply and demand chain and space chain. As early as 1937, Coase proposed that vertical integration depends on the size of transaction costs. To avoid opportunistic behaviors in transactions and reduce transaction costs, companies will choose vertical integration [24], and replace commodity contracts of transaction products with labor factor contracts [25].

Asset specificity, uncertainty, and transaction frequency are three key factors that affect transactions [26]. If the special assets invested in the transaction are misappropriated, the value of the assets will be reduced, and business operations will be affected [27]. When the specific assets of the company are relatively high, the operational losses can be reduced by adopting closer vertical cooperation. The theory of industry organization further points out that the key to vertical integration is maintaining the control of upstream and downstream businesses. Scope economy promotes the synergy of the industry chain, and downstream companies have strong monopoly power and greater control over the pricing power of upstream companies. Vertical integration expands the scale of companies [28], reduces the risk of opportunism by internalizing the trading market, increases the profits of the industry chain [29] and re-allows monopoly benefits [30].

When the degree of uncertainty and risk is high, diversified business operations can offset some of the risks and protect the long-term survival and development of the company. The real options theory points out that uncertainty is the source of income, not risk and threat [31]. Vertical integration can avoid market risks and reduce the uncertainty of company production and sales [27, 32].

The more frequent the transaction is, the more transaction costs would be. It is necessary to establish a closer collaboration method to reduce transaction costs. At the same time, the cost of establishing such a dedicated governance structure will be easily compensated [27]. From the perspective of endogenous competitive advantage, the theory of company capability believes that vertical integration can better combine complementary skills inside and outside the company, produce management and operating synergy and improve production efficiency [33].

Institutional environment theory analyzes the impact of vertical integration from the external institutional level of the industry chain. When the legal environment in the area where the company is located is better, the risk of default when the company signs a contract and the cost of contract execution will be reduced. When the environment and investor protections are poor, companies will merge the production originally organized by the market into the company, thus saving transaction costs and reducing corporate risks [34]. The uncertainty brought by changes in the policy environment affects the vertical integration of companies [35].

Based on the previous theoretical studies, we start from the on-chain and off-chain of the industry chain, with transaction cost theory and institutional environment theory as the main theoretical framework, focusing on the analysis of transaction costs and institutional environment on the vertical integration of pig industry chain while considering other factors (Fig 1).

## Vertical integration calculation and comparative analysis of hogpig industry chain

### The level of vertical integration

The I-O method (input-output method) is used to calculate the integration index of the subject of pig industry chain by using the market share data and the input-output relationship.

Indicators for the input-output method are constructed. According to the end-of-year and mid-year reports of the subject, among the business incomes classified by industry, the

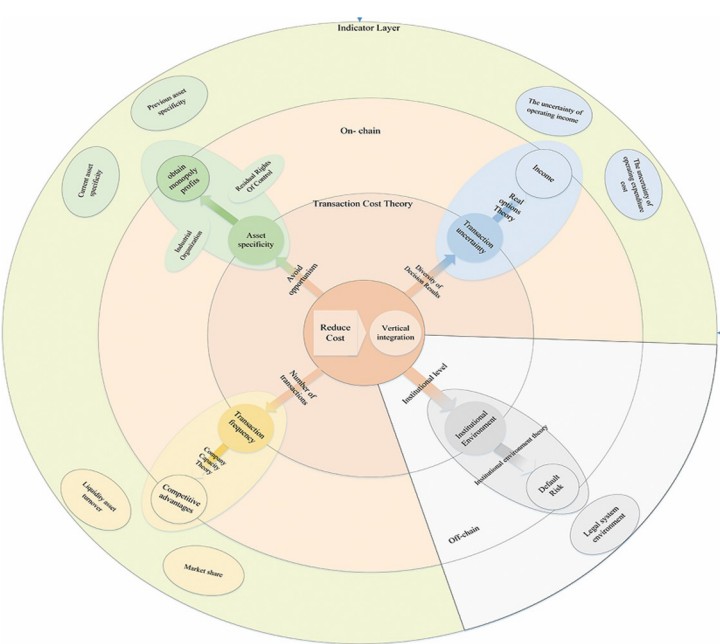

**Fig 1. Theoretical framework.**

industry with the highest proportion of business income is positioned as the main department $i$, and the other business income industries $j$ are the auxiliary department of the subject of pig industry chain. Through the input-output table issued by the National Bureau of Statistics in 2018, the input $\vartheta_{ij}$ of $j$ per unit output required for $i$ is calculated, as is the input $\vartheta_{ij}$ of $j$ per unit output inflow to $i$, $Mean_{ij,imput} = \frac{\vartheta_{ij}+\vartheta_{jt}}{2}$. This variable represents the integration opportunity of the main department $i$ and the auxiliary department $j$. $\gamma_j$ is the income ratio of the auxiliary department $j$ to the total income of the non-main department (Table 1).

Backward Integration Degree Model:

$$V_{backward} = \sum r_j\, Mean_{ij,imput} \tag{1}$$

Forward Integration Degree Model:

$$V_{forward} = \sum r_j\, Mean_{ij,imput} \tag{2}$$

## Data selection

According to the main business income, the companies with the highest operating income of pig-related products were selected. Based on this standard, there are 25 listed pig companies.

**Table 1. Statistical description.**

|  | Max. | Min. | Avg. |
|---|---|---|---|
| $Mean_{ij,imput}$ | 0.34 | 0.01 | 0.16 |
| $Mean_{ij,imput}$ | 0.34 | 0.01 | 0.17 |
| $Income_{total}$ | 72,526,310,239.06 | 214,104,119.26 | 13,464,032,767.95 |
| $Income_t$ | 57,557,623,514.21 | 221,178.26 | 2,880,825,559.80 |
| $Income_{forward\ j}$ | 22,577,165,714.29 | 2,475.00 | 991,398,663.71 |
| $Income_{backward\ j}$ | 1,594,575,386.58 | 12,688.27 | 203,505,802.52 |

To ensure the continuity of main business data from 2012 to 2019, 4 listed companies were selected for each link in the end. The annual reports and 2018 national input-output table of 12 listed pig companies were used as data sources. The listed pig companies are selected to cover the three links of feed, breeding, and slaughter in pig industry chain. With the form of "company + farmer", they lead farmers to connect to the market, gradually increase their market share, and form an industrial layout all over the country, covering more than 70% of pig breeding and more than 90% of pork supply. In Feed processing, the listed companies we selected were TRS, TKSW, ZBKJ, and JXN. Pig breeding was WSGF, MYGF, TBGF, CJNM, and Slaughter circulation was XWF, SHFZ, LDRS, and SXNY. Among them, 12 listed companies have more than 30 national-level core pig farms and more than 40 cooperative breeding.

## Analysis of the operating status of sample companies

At the present stage, the industrialization mode of pig farming in our country is still established on the basis of the household management. In 2019, the number of small-scale farms (households) with less than 500 pigs is still more than 99%, and small and medium-sized households are the absolute subject of the pig supply. Leading pig enterprises are required to participate in the transformation of pig industry to professional and modern breeding. In 2021, the market share of CR10 of pig enterprises is about 12.34% in China, and the scale enterprises have a very high boundary to expand. The leading enterprises of China's pig industry chain are rapidly expanding their scale and occupying the market of the production link through various forms of integration. Therefore, this paper first analyzes the selected sample companies (Table 2).

WSGF focuses on the Guangdong province and Guangxi province, the production capacity of MYGF is located in the area north of the Yangtze River, the market share of two companies has always been the top two in the industry. According to "2020 China Pig Industry Data Analysis report" collected by weiheng Agriculture, we found that Smithfield Foods companies is the world's leading sow-listed company with 1.24 million sows, WSGF, MYGF and ZBKJ ranked 2nd, 4th and 6th in the world with 1.2 million, 680,000 and 400,000 head, respectively.

TRS has developed the pig breeding business in Guangdong province and Hunan province, they produced 4000 thousand tons of feed in 2019.TKSW has 1.5 million tons of feed processing base and 1 million pig slaughtering and meat food processing base, the company produced 1330 thousand tons in 2019.The main business of ZBKJ is feed production and pig breeding. In 2019, the company produced 3,850 thousand tons of feed. The feed sector of JXN has covered

**Table 2. Descriptive analysis of sample companies (Unit: Ten thousand head).**

| Main business | Name | Location | Number of pigs in 2018 | Market share in 2019 | Number of pigs in 2019 | Market share in 2019 |
|---|---|---|---|---|---|---|
| **Pig breeding** | Wenshi(WSGF) | Guangdong | 2229.72 | 3.21% | 1856 | 3.41% |
| | Muyuan(MYGF) | Henan | 1101.2 | 1.59% | 1025 | 1.88% |
| | Tianbang(TBGF) | Zhejiang | 216.97 | 0.31% | 243.9 | 0.45% |
| | Chuanjiao(CJNM) | Sichuan | - | - | - | - |
| **Feed processing** | Tangrenshen(TRS) | Hunan | 68.1 | 0.10% | 83.9 | 0.15% |
| | Tiankang(TKSW) | Xinjiang | 64.66 | 0.09% | 84.3 | 0.15% |
| | Zhengbang(ZBKJ) | Jiangxi | 553.99 | 0.80% | 578.4 | 1.06% |
| | Jinxinnong(JXN) | Guangdong | 24.13 | 0.03% | 39.54 | 0.07% |
| **Slaughter circulation** | Xinwufeng(XWF) | Hunan | 74 | 0.11% | *64* | 0.12% |
| | Shuanghui(SHFZ) | Henan | 1630.56 (Slaughter pigs) | - | 1320 (Slaughter pigs) | - |
| | Longda(LDRS) | Shandong | 32.76 | 0.05% | 25.29 | 0.05% |
| | Shunxin(SXNY) | Beijing | 147.86 (Slaughter pigs) | - | 150 (Slaughter pigs) | - |

the research, production and marketing of pig feed in the whole department, and has built 12 large-scale production bases in Guangdong, Hunan, Hubei, Zhejiang, Anhui, Henan, Liaoning, Jilin, Heilongjiang and other places, with an annual capacity of more than 1 million tons.

SHFZ is a large food group mainly engaged in meat processing. It has built processing bases in 18 provinces in China, and slaughtering business accounts for more than 50% of its revenue. SXNY has always focused its pork sector on slaughtering business, with a complete industry chain integrating pig breeding, slaughtering and processing, deep processing of meat products and cold chain logistics distribution, it has promoted the distribution of breeding production capacity in the Beijing area. LDRS has built more than 20 food processing bases across the country, with a food processing capacity of 50,000 tons. The 12 listed companies can represent pig listed companies in China.

In summary, since the outbreak of ASF in China in 2018, the proportion of leading pig companies in the market has shown a trend of continuous growth, and the competition pattern of leading pig companies has gradually formed. Under the impact of ASF, the "head effect" has brought competitive advantages such as higher income, faster development speed and high attention from the market to pig enterprises, which has ushered in greater development opportunities for pig leading enterprises.

**Table 3. The degree of vertical integration in pig industry chain in the first half year, the second half, and the whole year 2012–2019 (%).**

| Year | | | 2012 | 2013 | 2014 | 2015 | 2016 | 2017 | 2018 | 2019 |
|---|---|---|---|---|---|---|---|---|---|---|
| **Feed processing** | **Backward** | H1 | - | - | - | 0.91 | 0.27 | 0.06 | 0.02 | 0.1 |
| | | H2 | - | - | - | 13.79 | 52.6 | 62.07 | 62.94 | 70.78 |
| | | yearly | 1.53 | 0.86 | 0.93 | 0.89 | 0.41 | 0.12 | 0.12 | 0.09 |
| | **Forward** | H1 | - | - | - | 39.94 | 30.96 | 56.12 | 54.82 | 60.43 |
| | | H2 | - | - | - | 0.87 | 0.46 | 0.17 | 0.21 | 0.08 |
| | | yearly | 31.99 | 40.75 | 42.05 | 34.68 | 43.86 | 61.55 | 59.76 | 67.51 |
| **Pig breeding** | **Backward** | H1 | - | - | - | 31.12 | 31.12 | 17.1 | 66.95 | 60.06 |
| | | H2 | - | - | - | 1.71 | 2.21 | 4.14 | 5.08 | 6.87 |
| | | yearly | 35.23 | 31.66 | 32.14 | 36.76 | 32.92 | 37.75 | 35.79 | 39.04 |
| | **Forward** | H1 | - | - | - | 1.72 | 2.92 | 1.07 | 9.21 | 3.42 |
| | | H2 | - | - | - | 18.45 | 17.44 | 16.22 | 22.23 | 38.29 |
| | | yearly | 0 | 0 | 1.5 | 1.78 | 3.81 | 4.84 | 5.8 | 5.34 |
| **Slaughter circulation** | **Backward** | H1 | - | - | - | 21.7 | 18.76 | 17.43 | 11.45 | 0.17 |
| | | H2 | - | - | - | 41.84 | 37.27 | 37.52 | 57.61 | 46.37 |
| | | yearly | 4.23 | 3.86 | 3.1 | 3.95 | 3.19 | 2.57 | 0.79 | 0.52 |
| | **Forward** | H1 | - | - | - | 37.84 | 38.02 | 41.1 | 39.1 | 56.92 |
| | | H2 | - | - | - | 21.13 | 17.97 | 15.34 | 8.22 | 12.14 |
| | | yearly | 31.72 | 31.63 | 35.91 | 35.34 | 35.12 | 35.23 | 42.6 | 41.45 |
| **Avg.** | **Backward** | H1 | - | - | - | 17.91 | 16.72 | 11.53 | 26.14 | 20.11 |
| | | H2 | - | - | - | 19.11 | 30.69 | 34.58 | 41.88 | 41.34 |
| | | yearly | 13.66 | 12.12 | 12.06 | 13.87 | 12.17 | 13.48 | 12.24 | 13.22 |
| | **Forward** | H1 | - | - | - | 26.5 | 23.97 | 32.76 | 34.38 | 40.26 |
| | | H2 | - | - | - | 13.48 | 11.96 | 10.58 | 10.22 | 16.84 |
| | | yearly | 21.23 | 24.13 | 26.4 | 23.93 | 27.6 | 33.87 | 36.05 | 38.1 |

Note: H1 indicates the first half year and H2 indicates the second half.

## Comparative analysis of vertical integration of pig industry chain

The vertical integration of pig industry chain in China showed an upward trend, and the degree of forward integration was higher than that of backward integration (Table 3). It means that under the guidance of relevant policies in China, pig industry chain continuously has been developing vertical integration of industry according to the characteristics of the development of the pig industry. Through this development model, the combination of upstream and downstream industries was realized to reduce the transaction cost of the subject of pig industry chain.

The growth rate of vertical integration of pig industry chain decreased in the early stage and increased in the later stage. The growth rate of vertical integration declined in the early stage of ASF. At the beginning of ASF outbreak, the epidemic spread rapidly in China, and the pig industry was vulnerable and became the highest risk industry. If any link in the industry chain is impacted, the whole pig industry will have an impact. A large number of pigs were killed, the loss of farmers was huge, the main business of slaughter circulation companies appeared to downturn, and the pig transfer policy greatly reduced the growth rate of vertical integration of slaughter and circulation links. The process of vertical integration is stagnant. Therefore, in the early days of ASF, the pig industry as a whole appeared to a downturn. The feed processing industry should prevent and control risks and accelerate the pace of vertical integration. The growth rate of vertical integration increased in the later period of ASF. With the implementation of epidemic prevention and control measures, the introduction of favorable policies for the pig industry, and the rise of pig prices, the market of the pig industry has rebounded. Under the new industrial development environment, the pig breeding industry has become a high-profit industry. As the upstream and downstream of the pig breeding industry, feed processing, and slaughter processing have accelerated the layout of pig breeding.

By comparing and analyzing the vertical integration degree of the feed processing link, pig breeding link, and slaughter circulation link in pig industry chain, it is found that vertical integration degree of each link is on the rise, and that of the feed processing link is the highest (Fig 2). Specifically, the degree of forward integration of the feed processing link is generally gradually improving, while the degree of backward integration is generally low, which indicates that the subject of pig industry chain in feed processing tends to expand the downstream industry.

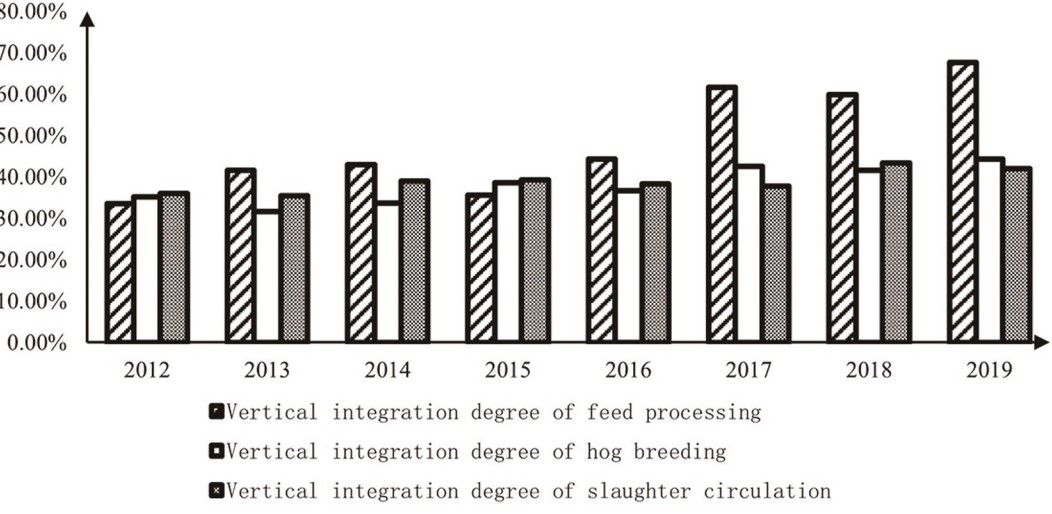

**Fig 2. The degree of vertical integration of each link in pig industry chain.**

The backward integration degree of pig breeding is higher than forward integration, which indicates that the subject of pig industry chain in pig breeding tends to expand the upstream industry. Forward integration degree of slaughter circulation is higher than backward integration, which indicates that the subject of pig industry chain in slaughter circulation tends to expand the downstream industry. The forward integration of feed processing and slaughter circulation is higher than backward integration. Feed processing link upstream only feeds the raw material industry, and the planting industry has the characteristics of high risk, high investment, and long period, so feed processing companies tend to forward integration. The feed industry in China has developed rapidly, the industry concentration has steadily improved, and the strength of companies has gradually increased. Existing leading companies have adopted the vertical integration development model. In contrast, the scale of pig breeding in China is insufficient, the concentration of the slaughtering and circulation industry is low for a long time, and industry vertical integration develops slowly. Slaughter circulation downstream only has a terminal retail industry. It has pricing power and high profit, so slaughtering and circulation companies tend to develop forward integration. Under the impact of ASF, the introduction of the pig welfare policy and the rising of pig prices increased the growth rate of vertical integration between feed processing and pig breeding (Fig 3).

## Materials and methods

### Variable selection

According to the above theoretical analysis, the variables selected in this paper are mainly divided into three parts.

**Transaction cost.** *(1) Asset specificity*. The asset specificity is positively correlated with the degree of vertical integration [36]. The assets of the pig industry are highly specialized, unable to be used for multiple purposes, and the capital conversion rate is very low. The subject of pig industry chain adopts a closer vertical integration of the industry chain to reduce operating losses when the asset specificity is high and ultimately improve the vertical integration of the subject of pig industry chain. previous asset specificity (Spe—1) was selected as specific indicators.

Asset specificity is expressed by the proportion of the current or previous fixed assets of the company in the total assets of the company

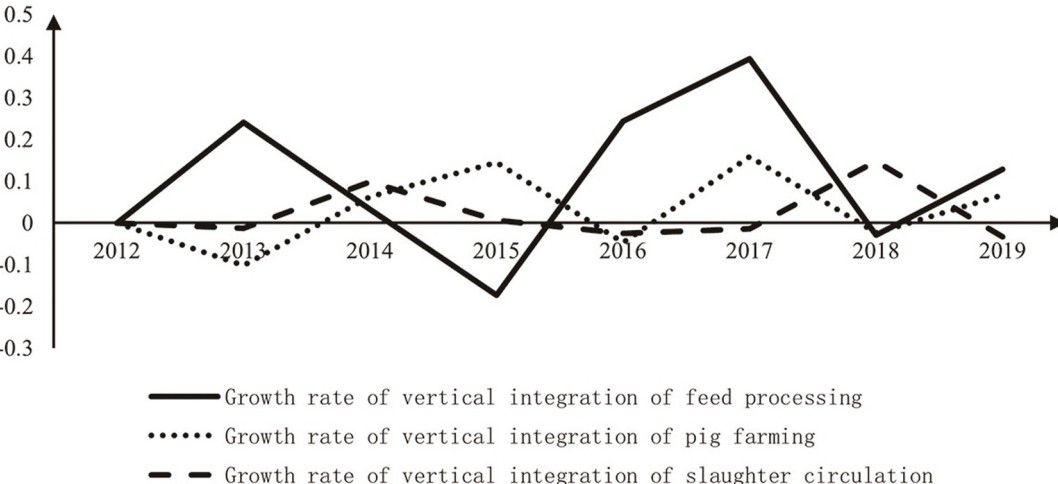

**Fig 3. The growth rate of the vertical integration of each link.**

1. Current asset specificity

$$Specificity = \frac{Fixedassets}{assets} \qquad (3)$$

2. Previous asset specificity

$$Specificity_{-1} = \frac{Fixedassets_{-1}}{assets_{-1}} \qquad (4)$$

*(2) Transaction uncertainty*. In the pig industry, the pork price fluctuates greatly, which makes many breeding subjects enter and exit frequently, leading to the uncertainty of the raw material supply of many slaughtering and processing subjects. Therefore, the higher the change rate of business income and the business cost is, the higher the transaction uncertainty would be, and so do the degree of vertical integration. The uncertainty of operating income (*sale*) and the uncertainty of operating expenditure cost (*cost*) are used.

$$Sale = \frac{income_t - income_{t-1}}{income_t} \qquad (5)$$

$$Cost = \frac{Cost_t - Cost_{t-1}}{Cost_t} \qquad (6)$$

$income_t$ is the main business income during the current period, $income_{t-1}$ is the main business income during the previous period, $cost_t$ is the main business cost during the current period, $cost_{t-1}$ is the main business cost during the previous period.

*(3) Transaction frequency*. Trading frequency refers to the number of transactions in a certain period. With each increase in the number of transactions, the relevant bargaining, signing, and other transaction costs increase a bit. In the production process of the pig in China, the transaction frequency of different links is different, resulting in different transaction costs and vertical integration of industry chains to different degrees.

1) Liquidity asset turnover (*Liquidity*). The higher turnover rate of current assets means a higher transaction frequency of the subject, which is towards vertical integration to reduce costs.

$$Liquidity = \frac{income_t * 2}{current_{begin} - current_{end}} \qquad (7)$$

$current_{begin}$ is the current assets at the beginning of the period, $current_{end}$ is the current assets at the end of the period.

2) Market share (*Market*). The higher the market demand for the main products of pig industry chain is, the more frequent the transaction with the upstream and downstream pig industry chain subjects would be. In addition, because the relationship between the industry chain is not optimized, the subject chooses upstream or downstream integration to control transaction costs when the transaction cost is high (Table 4). Market share is expressed by the main business income.

The market share variable is expressed by the ratio of the main business income of the company to the income of all companies in this link.

$$Market = Main\ business\ income \qquad (8)$$

Table 4. Variable descriptive statistics.

| Variable | Max. | Min. | Average |
|---|---|---|---|
| Explained variable | | | |
| $V_{forward}$ | 0.275 | 0 | 0.0255 |
| $V_{backward}$ | 0.3041 | 0 | 0.0808 |
| Explanatory variables | | | |
| Spe | 0.5421 | 0.1052 | 0.3186 |
| Spe-1 | 0.5421 | 0.1222 | 0.3336 |
| Sale | 1.334 | -0.362 | 0.1786 |
| Cost | 1.2897 | -0.3802 | 0.1683 |
| System | 24.03 | 2.03 | 8.8794 |
| Size | 1.9516 | -0.3656 | 0.1167 |
| Age | 22 | 1 | 9.7458 |
| Liquidity | 6.0573 | 0.1624 | 2.1088 |
| Market | 0.4312 | 0.0013 | 0.0847 |

**Legal system environment.** The better the legal system environment (*System*) is, the lower the degree of vertical integration is. The worse the regional legal system environment is, the greater the default risk faced by the subject of pig industry chain is, and the higher the transaction cost and bargaining cost are. Therefore, when the legal environment is poor, the subject merges the production originally organized by the market into pig industry chain to save transaction costs and reduce the risk. We used the legal environment index of the province where the sample company is registered. The specific index source from 2012 to 2018 is the Marketization Index Report in China by Province (2018).

**Control variable.** Company ability. The ability of pig companies influences vertical integration by affecting the competitive advantage of the subject. The characteristics of subjects, production and operation characteristics, and production scale also affect their behavioral choices and cause transaction uncertainty. When the subject has scarce and low alternative resources and capabilities, they choose to extend it to the relevant business areas through vertical integration to obtain profits. Company ability includes company size (*Size*)and company age (Age). The former is expressed by the change rate of the total employees of the company, and the latter is expressed by the establishment time of the company as of 2019.

$$Age = 2019 - Established + 1 \qquad (9)$$

$$Size = \frac{employes_t - employes_{t-1}}{employes_{t-1}} \qquad (10)$$

To prevent the correlation between explanatory variables and make the results deviate, SPSS was used to test the multicollinearity of the above explanatory variables. According to Table 5, VIF<0.1, tolerance > 0.1 condition index < 30, it can be seen that the explanatory variables do not have multiple collinearities and can be further regression analyzed.

## Model design

Based on the previous theoretical analysis, the breakpoint regression and Tobit model were used to estimate the impact of ASF on the vertical integration of pig industry chain.

**Breakpoint regression model.** At the beginning of 2018, ASF broke out. Assuming that the outbreak time is set as a breakpoint, the indicators before and after the outbreak are

**Table 5. Explanatory variable multicollinearity test.**

| Model | Forward | | | Backward | | |
|---|---|---|---|---|---|---|
| Variable | tolerance | VIF | condition index | tolerance | VIF | condition index |
| Cons. | | | 1 | | | 1 |
| Spe | 0.247 | 4.055 | 2.246 | 0.247 | 4.055 | 2.246 |
| Spe-1 | 0.216 | 4.622 | 2.791 | 0.216 | 4.622 | 2.791 |
| Sale | 0.223 | 4.474 | 3.246 | 0.223 | 4.474 | 3.246 |
| Cost | 0.239 | 4.181 | 5.03 | 0.239 | 4.181 | 5.03 |
| Liquidity | 0.502 | 1.994 | 5.278 | 0.502 | 1.994 | 5.278 |
| System | 0.451 | 2.216 | 8.267 | 0.451 | 2.216 | 8.267 |
| Market | 0.639 | 1.565 | 13.551 | 0.639 | 1.565 | 13.551 |
| Size | 0.875 | 1.143 | 20.652 | 0.875 | 1.143 | 20.652 |
| Age | 0.555 | 1.802 | 23.783 | 0.555 | 1.802 | 23.783 |

statistically measured by breakpoint regression analysis, and the applicability of the model is judged according to the actual situation. If the breakpoint of the model exists significantly: the vertical integration degree of pig industry chain jumps before and after the outbreak, it indicates that ASF has a greater impact on the vertical integration degree of pig industry chain. Therefore, this paper uses the breakpoint regression model to conduct an empirical study on the statistical measurement of the impact of ASF on the vertical integration of pig industry chain.

Set the breakpoint regression model as follows:

$$Y_t = \alpha + \beta D_t + G(X_t) + \varepsilon_t \tag{11}$$

$Y_i$ is the result variable, the subscript $\iota$ represents the $\iota$th sample, $G(X_t)$ represents a variety of factors that affect the result variable, $\varepsilon_t$ is a random disturbance term, and $D_i$ is the treatment variable that indicates whether it is affected by ASF, which can be expressed as Eq (12)

$$D_t = \begin{cases} 0, & IF\iota \leq 0 \\ 1, & IF\iota \geq 0 \end{cases} \tag{12}$$

The formula shows that the processing variable $D_t$ is a non-continuous function affected by ASF. The breakpoint regression model is used to estimate the effect of processing variable $D_t$ on the result variable $Y_t$. In this paper, the data on the integration degree of pig industry chain before 2018 are used as the control group, and the data after 2018 are used as the experimental group. In the observation sample, 0 is taken as the dividing point ($D_t = 1$, indicating that the result variable is affected. $D_t = 0$ indicates that the result variable is not affected.)

**Basic models of forward integration and backward integration.** Since the vertical integration degree calculated above is a latent variable with continuous distribution from 0 to 1, and the data distribution characteristics meet the censored feature, Tobit model was used for regression analysis in this paper.

$$V_{forward} = \partial_0 + \partial_1 Spe_{-1} + \partial_2 sale + \partial_3 cost + \partial_4 Frequency + \partial_5 System + \partial_6 Market + \partial_7 Size + \partial_8 Age + \sigma_1 \tag{13}$$

$$V_{backward} = \partial_0 + \partial_1 Spe_{-1} + \partial_2 sale + \partial_3 cost + \partial_4 Frequency + \partial_5 System + \partial_6 Market + \partial_7 Size + \partial_8 Age + \sigma_2 \tag{14}$$

## Results

### Breakpoint regression results

The breakpoint regression showed that there was a breakpoint in the backward integration of pig industry chain. The estimation of lwald200 was significantly negative at a 10% level, indicating that ASF significantly inhibited the backward integration, and the backward integration showed a significant downward jump after the occurrence of ASF (Fig 4). From Table 5, the estimation of lwald50 was significantly positive at the 1% level, indicating that ASF significantly promoted the forward integration of pig industry chain. The forward integration has an obvious upward jump after ASF (Fig 5). ASF has accelerated the forward integration of the pig industry in China.

The breakpoint regression shows that there is a breakpoint in the model test of the backward integration of pig industry chain affected by ASF (Table 6).

Breakpoint regression was used to verify the backward and forward integration degree of pig industry chain. The findings showed that the indicators meet the model assumptions, which are consistent with the trend of economic phenomena and can capture the characteristics of cliff-like changes in the vertical integration of pig industry chain.

### Empirical results of vertical integration of pig industry chain

The backward integration of pig industry chain results showed that the establishment time of the subject had a significant negative correlation, and the market demand was positively correlated before the outbreak of ASF (Table 7).

Compared with before the ASF outbreaks, the determinants of backward integration have noticeable change. First of all, the uncertainty of cost, establishment time, and size of the subject had a significant negative correlation. After ASF, the cost has become an important factor for the subject to consider whether to take vertical integration, the older and larger the subject, the more cost concerned. And the higher change rate of operating income means that the main production and operation of the pig industry chain are unstable, and the pig industry chain inclined to reduce forward integration. Secondly, transaction frequency had a positive

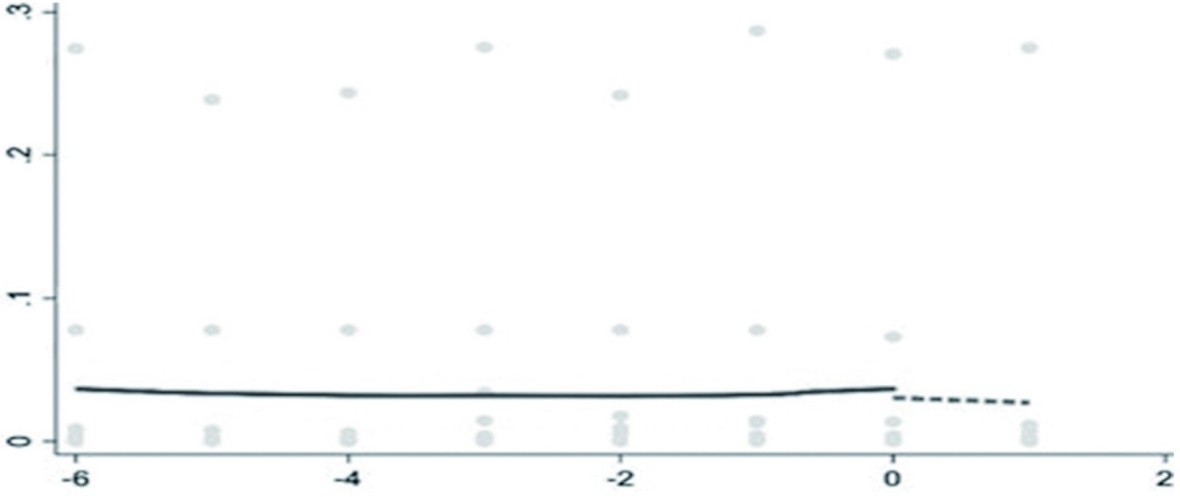

**Fig 4. The breakpoint regression of the backward integration graph.**

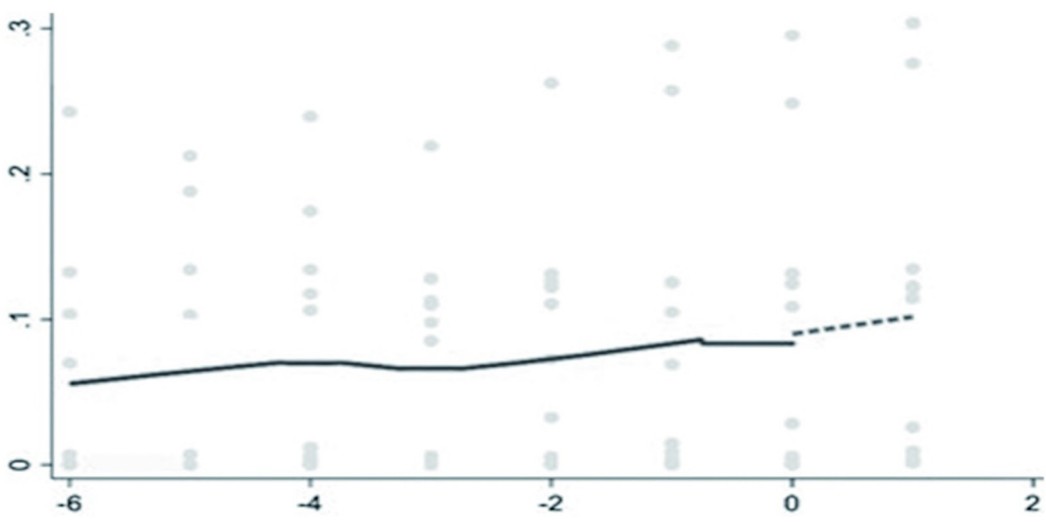

**Fig 5. The breakpoint regression of forward integration graph.**

correlation. The higher the transaction frequency is, the higher the degree of backward integration is.

In the forward integration of the pig industry chain, asset specificity of the previous period, legal system environment, and transaction frequency had a positive correlation, and the market share and company size were significantly negatively correlated before the outbreak of ASF (Table 8).

Compared with before the ASF outbreaks, the number of determinants of forward integration have no noticeable change. First of all, previous asset specificity and transaction frequency had improved the promotion effect on the backward integration degree of the pig industry chain, and the effect of the legal system environment changed little. The higher asset specificity means the subject had invested in the more proprietary assets, and the higher legal system environment means the subject had a better market trading environment and avoided the cost of default for the subject of the pig industry chain, the subject choose to develop and increase the frequency of transactions rapidly. Therefore, the degree of forward integration of companies will be higher. Secondly, the weakening effect of market demand on forward integration has been mitigated. Along with the enhancement of market demand, the scale of the product processing link of the industry chain subject gradually enlarged. When the processing link reaches the scale economy effect, this link would be separated from the original industry chain subject to improve the specialization degree of the industry chain. Therefore, the forward integration of the industry chain decreased with the increase in market demand. ASF has hit the hog industry hard, reducing market demand and industry development, so the weakening effect on forward integration is mitigated.

Because the pig slaughtering and processing link and pig breeding link are highly related, belong to the upstream and downstream pillar link. Therefore, we further analyze the determinants of vertical integration between breeding and slaughtering companies. Through empirical

**Table 6. The breakpoint regression of forward integration and backward integration result.**

| Result | | Coef. | Std. Err. | z | P>z |
|---|---|---|---|---|---|
| V forward | lwald50 | 0.0001197 | 0.0440993 | 0 | 0.998 |
| V backward | lwald200 | -0.0019852 | 0.0195356 | -0.1 | 0.919 |

**Table 7. The analysis results of the backward integration.**

| $V_{backward}$ | Model 1 | Model 2 |
|---|---|---|
| Spe-1 | -0.02 (0.04) | 0.04 (0.032) |
| Sale | -0.029 (0.04) | 0.02 (0.015) |
| Cost | -0.003 (0.036) | -0.04* (0.017) |
| Liquidity | 0.019 (0.012) | 0.014* (0.007) |
| System | 0.0036 (0.005) | 0.002 (0.002) |
| Market | 0.097* (0.053) | -0.068 (0.048) |
| Size | -0.002 (0.02) | -0.260*** (0.07) |
| Age | -0.011* (0.006) | -0.016*** (0.004) |
| Constant | 0.079 (0.108) | 0.405*** (0.086) |

Note: (1) Model 1 is the regression result for 2012–2017 and model 2 is the regression result for 2018–2019.

(2) *** $p < 0.01$,

** $p < 0.05$,

* $p < 0.1$.

analysis, it is found that after the outbreak of ASF, the determinants of backward integration of the two types of enterprises do not change significantly, and the previous asset specificity has the greatest impact on the two types of companies. The determinants of forward integration of the two types of companies have increased, and previous asset specificity still has the largest effect (Table 9).

## Discussion

This paper studies the vertical integration of pig industry chain from the perspective of stable supply. The study found that the degree of vertical integration of pig industry chain has been increasing year by year in China. The outbreak of ASF has disrupted the operation of the feed, breeding and slaughtering, and processing links in pig industry chain. The feed processing link pays more attention to forward integration. Under the guidance of the actual market demand, feed companies choose to use feed with high digestion and high disease resistance to ensure the safety of pig breeding and reduce production costs. The degree of backward

**Table 8. The analysis results of the forward integration.**

| $V_{forward}$ | Model 1 | Model 2 |
|---|---|---|
| Spe-1 | 0.124*** (0.024) | 0.16*** (0.015) |
| Sale | 0.001 (0.02) | 0.004 (0.007) |
| Cost | -0.006 (0.02) | -0.10 (0.007) |
| Liquidity | 0.019** (0.007) | 0.03*** (0.004) |
| System | 0.02*** (0.003) | 0.018*** (0.001) |
| Market | -0.136*** (0.046) | -0.124*** (0.0219) |
| Size | -0.021 (0.02) | -0.02 (0.04) |
| Age | -0.012** (0.004) | -0.01*** (0.002) |
| Constant | 0.19** (0.07) | 0.24*** (0.05) |

Note: (1) Model 1 is the regression result for 2012–2017 and model 2 is the regression result for 2018–2019.

(2) *** $p < 0.01$,

** $p < 0.05$,

* $p < 0.1$.

**Table 9. The analysis results of slaughtering link and breeding link.**

| Variable | 2012–2017 | | 2018–2019 | |
| --- | --- | --- | --- | --- |
| | $V_{forward}$ | $V_{backward}$ | $V_{forward}$ | $V_{backward}$ |
| Spe-1 | 0.192*** (0.025) | 0.06 (0.05) | 0.09*** (0.02) | 0.076* (0.032) |
| Sale | -0.016 (0.015) | -0.01 (0.03) | -0.02** (0.007) | 0.03* (0.013) |
| Cost | -0.001 (0.017) | -0.02 (0.03) | 0.0170* (0.007) | -0.058** (0.017) |
| Liquidity | 0.04*** (0.0096) | 0.015 (0.014) | 0.0494*** (0.004) | 0.0311* (0.012) |
| System | 0.02*** (0.0022) | 0.003 (0.005) | 0.00918*** (0.002) | 0.007 (0.004) |
| Market | -0.23*** (0.046) | 0.016 (0.048) | -0.308*** (0.039) | -0.06 (0.08) |
| Size | -0.037 (0.026) | -0.005 (0.02) | 0.0987** (0.039) | -0.31*** (0.0611) |
| Age | -0.00406 (0.005) | -0.01 (0.007) | 0.0108* (0.005) | -0.024* (0.01) |
| Constant | 0.102 (0.086) | 0.22 (0.134) | -0.211* (0.095) | -0.211* (0.10) |

*** $p < 0.01$,

** $p < 0.05$,

* $p < 0.1$.

integration in the breeding link is higher than forward integration. Listed companies have large-scale farming, and their marginal cost elasticity is small. They will choose more to cooperate with upstream raw material suppliers, and widen the gap with the industry average cost. In the context of the policy of prohibiting the inter-provincial transfer of pigs, slaughtering companies are more likely to choose forward integration to avoid overstocking and ensure local pork supply.

ASF has promoted the forward integration of the pig industry chain in China, which results from the fact that forward integration enables companies to better control the retail price and to respond more effectively to the changes in market demand changes [37]. According to the transaction cost theory, transaction cost, transaction frequency, and transaction uncertainty will affect the way companies choose vertical integration. Therefore, this paper further explores the determinants that affect the adoption of forward and backward integration by listed pig companies. The empirical results show that previous asset specificity, transaction frequency, legal system environment, and market demand have a significant correlation with the forward integration of the pig industry chain under the impact of ASF. It is also confirmed that due to the impact of the epidemic, the price of pigs fluctuates greatly, consumer doubts arise, and market demand is weak. In this context, pig industry chain is suitable for forward integration. Policy support for the forward integration of pig industry chain in areas with relatively low legal and institutional environments should be increased.

This study also has several limitations. Firstly, the research time is limited. In this paper, we study the sample deadline for 2019, and the reason in the following aspects. First of all, the impact of the epidemic on the pig industry is different. In 2018, the African swine fever outbreak has produced a huge shock on pig production in China, it brought a great shock on the pig supply, further affecting the whole industry chain vertical integration. The impact of the COVID-19 outbreak in January 2020 on China's pig industry is mainly reflected in the demand side. Secondly, the impact of vertical integration of China's pig industry chain is mainly due to African swine fever, and the impact of COVID-19 will eventually dissipate. In the late period of the COVID-19 epidemic, with the gradual rebound of pork consumption by domestic residents, the impact on the pig industry has gradually weakened, and the stable supply of pigs is still the development goal of the pig industry. To make the research focus more specific, this paper takes ASF as the research object and does not include the years that cover

the impact of COVID-19, ensuring that the research focuses on the supply side of China's pig industry.

## Conclusions

After the outbreak of ASF, pig industry and pork supply in China was facing new development challenges. Under the normalization of ASF, vertical integration can reduce transaction costs of pig companies, and stabilize supply in pig industry chain, which has become an important way to ensure a stable supply of pigs. The main conclusions are as follows:

1. The degree of vertical integration of the pig industry chain in China has been increasing year by year. Influenced by the ASF epidemic, the degree of vertical integration of the subject in different links fluctuates in different situations. The backward integration of the pig breeding link is higher than the forward integration, and the forward integration of the feed processing link and slaughter circulation link is higher than the backward integration. The integration of feed and breeding links to a certain extent reflects the integration of the pig industry within the primary industry. The vertical integration method based on the combination of planting and breeding not only ensures food safety but also promotes the sustainable development of the pig industry. ASF has promoted changes in domestic live pig transportation policies, and the transformation of live pig transportation into chilled meat transportation has stimulated forward integration in the slaughter circulation link.

2. Under the impact of ASF, the vertical integration degree of pig industry chain jump, while ASF inhibits the backward integration degree of pig industry chain and promotes forward integration. Pig industry chain is taking the forward integration, increasing the supply of pork by improving the control over the supply of upstream raw materials.

3. Under the impact of ASF, the determinants of the backward integration have increased, such as uncertainty of cost, transaction frequency, and company size. These factors have a significant impact on the backward integration of the pig industry chain. the higher rate of change in operating cost means that the production and operation of the pig industry chain are unstable, and the pig industry chain inclined to carry out the forward integration.

4. Under the impact of ASF, the number of determinants of forward integration has not changed significantly. A higher legal system environment means the subject had a better market trading environment. As the subject increases the input of dedicated assets, transaction frequency rises and forward integration can increase. The increase in market demand will increase the scale of product processing link for the subject. When the processing link reaches economies of scale, this link will be separated from the subject of the original industry chain, and increase the specialization of the companies. Therefore, the degree of forward integration will decrease with the increase in market share. Therefore, pig companies can choose to improve the proprietary assets, increase the transaction frequency, and maintain a stable legal environment to promote the forward integration of the industry chain and stabilize the supply of pigs.

## Supporting information

**S1 Table. Influence factors.**
(XLSX)

## Acknowledgments

We would like to thank the editors and two reviewers whose comments helped to greatly improve this manuscript.

## Author Contributions

**Conceptualization:** Gangyi Wang.

**Data curation:** Siyu Chen, Chang´e Zhao.

**Formal analysis:** Jingjing Wang, Siyu Chen.

**Funding acquisition:** Gangyi Wang.

**Methodology:** Jingjing Wang.

**Resources:** Siyu Chen.

**Software:** Jingjing Wang.

**Validation:** Chang´e Zhao.

**Writing – original draft:** Jingjing Wang, Siyu Chen.

**Writing – review & editing:** Gangyi Wang, Jingjing Wang.

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
