## [Decision Letter · Decision Letter 0]

7 Sep 2022

PONE-D-22-21687Vertical Integration Selection of Chinese Hog Industry Chain Under African Swine Fever - From the Perspective of Stable Hog SupplyPLOS ONE

Dear Dr. wang,

Thank you for submitting your manuscript to PLOS ONE. After careful consideration, we feel that it has merit but does not fully meet PLOS ONE’s publication criteria as it currently stands. Therefore, we invite you to submit a revised version of the manuscript that addresses the points raised during the review process. Please submit your revised manuscript by Oct 22 2022 11:59PM. If you will need more time than this to complete your revisions, please reply to this message or contact the journal office at plosone@plos.org. Please include the following items when submitting your revised manuscript:A rebuttal letter that responds to each point raised by the academic editor and reviewer(s). You should upload this letter as a separate file labeled 'Response to Reviewers'.A marked-up copy of your manuscript that highlights changes made to the original version. You should upload this as a separate file labeled 'Revised Manuscript with Track Changes'.An unmarked version of your revised paper without tracked changes. You should upload this as a separate file labeled 'Manuscript'.

We look forward to receiving your revised manuscript.

Kind regards,

Douglas Gladue, Ph.D

Academic Editor

PLOS ONE

Journal Requirements:

4. Please ensure that you refer to Figures 1, in your text as, if accepted, production will need this reference to link the reader to the figure.

5. We note you have included a table to which you do not refer in the text of your manuscript. Please ensure that you refer to Table 1 in your text; if accepted, production will need this reference to link the reader to the Table.

Additional Editor Comments:

I was only able to secure one reviewer, but upon examination of the manuscript I agree with the reviewer, please address all the concerns of the reviewer upon resubmission.

Reviewers' comments:

Reviewer's Responses to Questions

**Comments to the Author**

1. Is the manuscript technically sound, and do the data support the conclusions?

Reviewer #1: Partly

2. Has the statistical analysis been performed appropriately and rigorously? 

Reviewer #1: I Don't Know

3. Have the authors made all data underlying the findings in their manuscript fully available?

Reviewer #1: No

4. Is the manuscript presented in an intelligible fashion and written in standard English?

Reviewer #1: No

5. Review Comments to the Author

Reviewer #1: The present study analyzed the driving forces behind the vertical integration of the listed pig companies in China in response to African swine fever outbreaks. The work is helpful for the understanding of the recent dramatic changes of the Chinese pig Industry. However, several concerns need to be addressed. The data from the latest years of the 12 listed Chinese pig companies should be included in the analysis by more independent methods. The manuscript was poorly written and should be revised by native English speakers and agro-economists.

6. PLOS authors have the option to publish the peer review history of their article (what does this mean?). If published, this will include your full peer review and any attached files.

Reviewer #1: **Yes: **Hua-Ji Qiu

---

## [Author Response · Author response to Decision Letter 0]

25 Oct 2022

Dear editors and reviewers,

Thank you for your valuable feedback on our manuscript entitled " Vertical Integration Selection of Chinese Pig Industry Chain under African Swine Fever - From the Perspective of Stable Pig Supply ". Those comments are all valuable and very helpful for revising and improving our paper, as well as the important guiding significance to our research.

We have carefully studied the comments of the editor and reviewer comments, and based on the comments, the paper has been modified as follows (the red font below is the comments of the editor and reviewers, and the black font below is the modification description):

Responses to the comments of the editors

Comment-1: Please ensure that your manuscript meets PLOS ONE's style requirements, including those for file naming. 

Response: Thank you sincerely for this comment. We can ensure that our manuscript meets PLOS ONE's style requirements.

Comment-2: We note that the grant information you provided in the ‘Funding Information’ and ‘Financial Disclosure’ sections do not match. When you resubmit, please ensure that you provide the correct grant numbers for the awards you received for your study in the ‘Funding Information’ section.

Response: Thank you sincerely for this comment. Based on the comment of the editor, our cover letter includes the updated financial statement. Details are as follows: 'This research is supported by the Natural Science Foundation of Heilongjiang Province China (Project number LH2019G002); and the Humanities and Social Science Foundation of Ministry of education of China (Project number 21YJA790053). The funders had no role in study design, data collection and analysis, decision to publish, or preparation of the manuscript. '

Comment-3: In your Data Availability statement, you have not specified where the minimal data set underlying the results described in your manuscript can be found. PLOS defines a study's minimal data set as the underlying data used to reach the conclusions drawn in the manuscript and any additional data required to replicate the reported study findings in their entirety. All PLOS journals require that the minimal data set be made fully available. For more information about our data policy, please see http://journals.plos.org/plosone/s/data-availability.

Response: Thank you sincerely for this comment. We have added “S1 Table. Influence factors.xlsx” as Supporting information

Comment-4: Please ensure that you refer to Figures 1, in your text as, if accepted, production will need this reference to link the reader to the figure.

Response: Thank you sincerely for this comment. We have referred to Figures 1 in our text.

Comment-5: We note you have included a table to which you do not refer in the text of your manuscript. Please ensure that you refer to Table 1 in your text; if accepted, production will need this reference to link the reader to the Table.

Response: Thank you sincerely for this comment. We have referred to Table 1 in our text.

Responses to the comments of Reviewer #1

Comment: The present study analyzed the driving forces behind the vertical integration of the listed pig companies in China in response to African swine fever outbreaks. The work is helpful for the understanding of the recent dramatic changes of the Chinese pig Industry. However, several concerns need to be addressed. The data from the latest years of the 12 listed Chinese pig companies should be included in the analysis by more independent methods. The manuscript was poorly written and should be revised by native English speakers and agro-economists.

 Response: Thank you sincerely for this comment. Based on the comments of the reviewers, first of all, we added the analysis of the sample companies(Line 214-248), secondly, we added the discussion on the determinants of vertical integration of enterprises in different links in the empirical analysis section(Line456-466 ), finally, we have optimized the grammar and language.

With best regards

Yours sincerely,

Gangyi Wang

Jingjing Wang

Siyu Chen

Chang’e Zhao

---

## [Decision Letter · Decision Letter 1]

5 Jan 2023

Vertical Integration Selection of Chinese Pig Industry Chain under African Swine Fever - From the Perspective of Stable Pig Supply

PONE-D-22-21687R1

Dear Dr. wang,

We’re pleased to inform you that your manuscript has been judged scientifically suitable for publication and will be formally accepted for publication once it meets all outstanding technical requirements.

Kind regards,

Douglas Gladue, Ph.D

Academic Editor

PLOS ONE

Additional Editor Comments (optional):

Reviewers' comments:

Reviewer's Responses to Questions

**Comments to the Author**

1. If the authors have adequately addressed your comments raised in a previous round of review and you feel that this manuscript is now acceptable for publication, you may indicate that here to bypass the “Comments to the Author” section, enter your conflict of interest statement in the “Confidential to Editor” section, and submit your "Accept" recommendation.

Reviewer #1: All comments have been addressed

2. Is the manuscript technically sound, and do the data support the conclusions?

Reviewer #1: Yes

3. Has the statistical analysis been performed appropriately and rigorously? 

Reviewer #1: Yes

4. Have the authors made all data underlying the findings in their manuscript fully available?

Reviewer #1: Yes

5. Is the manuscript presented in an intelligible fashion and written in standard English?

Reviewer #1: (No Response)

6. Review Comments to the Author

Reviewer #1: (No Response)

7. PLOS authors have the option to publish the peer review history of their article (what does this mean?). If published, this will include your full peer review and any attached files.

Reviewer #1: **Yes: **Hua-Ji Qiu

---

## [Editor Report · Acceptance letter]

10 Feb 2023

PONE-D-22-21687R1 

Vertical Integration Selection of Chinese Pig Industry Chain under African Swine Fever - From the Perspective of Stable Pig Supply 

Dear Dr. Wang:

I'm pleased to inform you that your manuscript has been deemed suitable for publication in PLOS ONE. Congratulations! Your manuscript is now with our production department. 

Kind regards, 

on behalf of

Dr. Douglas Gladue 

Academic Editor

PLOS ONE